# OPTIMISATION-BASED MULTI-MODAL SEMANTIC IMAGE EDITING

## ABSTRACT

Image editing affords increased control over the aesthetics and content of generated images. Pre-existing works focus predominantly on text-based instructions to achieve desired image modifications, which limit edit precision and accuracy. In this work, we propose an inference-time editing optimisation, designed to extend beyond textual edits to accommodate multiple editing instruction types (*e.g.* spatial layout-based; pose, scribbles, edge maps). We propose to disentangle the editing task into two competing subtasks: successful local image modifications and global content consistency preservation, where subtasks are guided through two dedicated loss functions. By allowing to adjust the influence of each loss function, we build a flexible editing solution that can be adjusted to user preferences. We evaluate our method using text, pose and scribble edit conditions, and highlight our ability to achieve complex edits, through both qualitative and quantitative experiments.

## 1 INTRODUCTION

Large scale text-to-image generative diffusion models have revolutionised image generation capabilities. Recent foundation models such as DALL-E (Ramesh et al., 2022), Stable Diffusion (Rombach et al., 2022) and Imagen (Saharia et al., 2022) have achieved impressive results in terms of composition and image quality. While these models successfully leverage the power of text conditioning, precise controllability of image outputs remain limited and achieving desired outcomes often requires time-consuming and cumbersome prompt engineering (Sonkar et al., 2022). A natural way of increasing control over image structure and simplifying this process is to introduce additional conditioning in the form of image layout descriptions. Recent works have introduced additional trainable modules that, when combined with a pre-trained foundation model, allow to condition generated images according to specific layout instructions. Such conditioning has been realised through object bounding boxes (Li et al., 2023), poses, edge maps, scribbles and segmentation maps (Zhang & Agrawala, 2023; Mou et al., 2023). The increased controllability provided by these modules have further highlighted a key missing component: editability of generated images. A small modification to layout or text instructions can lead to large image modifications, in particular for looser constraints (e.g. pose). Instead of iterative prompt updates with no guarantees of maintaining a consistent image appearance, editing can afford to locally modify an image while consistently preserving desired aesthetics.

In light of these potential benefits, text-driven image editing has been heavily explored recently. With the exception of large models with intrinsic editing properties (Balaji et al., 2022; Chang et al., 2023), editing solutions typically leverage a pre-trained foundational model and introduce a novel editing mechanism. Methods can be separated in three categories: 1) training-based methods that fine-tune the base models (Brooks et al., 2023; Zhang et al., 2023; Kawar et al., 2023), 2) training-free approaches that keep the generative model frozen (Hertz et al., 2022; Couairon et al., 2022b; Tumanyan et al., 2023) and 3) inference-time optimisation (Ma et al., 2023; Parmar et al., 2023; Epstein et al., 2023; Dong et al., 2023). Training-based approaches achieve higher quality edits, at the cost of expensive dataset construction, training and reduced generalisability. Training-free editing typically relies on attention manipulation or inversion mechanisms. These methods are faster and more practical, but also parameter sensitive and more restricted in editing ability. Alternatively, one can leverage ideas from both lines of research by considering inference-time optimisation, *i.e.* optimise intermediate image features directly instead of model weights. This strategy increases control over the edit process, while maintaining the flexibility of training free methods. Inference-time optimisation has been

leveraged to guide cross attention mechanisms (Ma et al., 2023; Parmar et al., 2023), move and modify instances through features and attention updates (Epstein et al., 2023) and automatically tune input prompts (Dong et al., 2023); increasing control over the structure of generated images.

Nonetheless, the aforementioned methods mainly rely on text-driven instructions to update image content. As discussed above, this can limit the precision and accuracy of required edits, as well as introduce high sensitivity and brittleness to the considered text instructions. In this work, we propose a novel editing method that relies on inference-time optimisation of latent image features that is intrinsically designed to work with a variety of conditioning signals, including text, pose or scribble instructions. We disentangle the editing task into preservation and modification tasks via two optimisation losses. The former is realised by imposing consistency between latent features of the input and edited image, while the latter leverages an intermediate output (the *guidance image*) with relaxed preservation constraints to guide the editing process in target modification areas. Our method can make use of layout control modules, effectively allowing editing using a wide range of editing queries. By disentangling preservation and modification tasks, we provide a flexible solution where the influence of each subtask can be adjusted to user preferences. We evaluate our method using text, pose and scribble conditions, We highlight our ability to achieve complex edits, from multiple conditions, and evaluate performance using both qualitative and quantitative experiments.

To summarise, our main contributions are the following:

- A novel image editing method for frozen diffusion models that goes beyond text editing, additionally capable of handling image layout conditions.
- A disentangled inference-time optimisation-based editing strategy that separates background preservation from foreground editing, affording flexibility to increase focus on one subtask.
- An adaptive solution that can adjust the editing process to task complexity.

## 2 RELATED WORK

In this section, we review closely related work on image editing, with an emphasis on text-guided editing under text-to-image diffusion models. We identify three main methodologies: fine-tuning based, training-free and inference-time optimisation editing methods.

**Fine-tuning based image editing** enables a pre-trained diffusion model to edit images by fine-tuning model weights. InstructPix2Pix (Brooks et al., 2023) leverages a large language model to build an extensive training dataset comprising image, edit-instruction pairs that are used to fine-tune a diffusion model to achieve editing based on written instructions. Imagic (Kawar et al., 2023) fine-tunes the textual-embedding and the generative model to ensure high reconstruction ability on each specific image, then interpolates between the original and editing prompt in order to obtain the edited image. Lastly, SINE (Zhang et al., 2023) fine-tunes a pre-trained models on each image to edit, then combines predictions from the pre-trained and fine-tuned models, conditioned on the edit and original image prompt respectively. These techniques generally require building a large training dataset, or per image fine-tuning at inference time. Both constraints can be costly, and notably make extensions to additional types of edit conditions (beyond text) more complex and expensive.

**Training-free image editing** achieves image modification using a frozen pre-trained model. These approaches typically involve manipulation of the denoising diffusion process to introduce editing functionality. Prompt-to-prompt (Hertz et al., 2022) enables image editing by manipulating cross-attention maps between text and image features and exploiting their differences when considering the original and edit prompts. Similarly, Plug and play (Tumanyan et al., 2023) leverages attention mechanisms, injecting self attention features from the original image into the generation process of the edited image. Ravi et al. (2023) propose an editing method, built on top of the DALL-E 2 model (Ramesh et al., 2022), that leverages CLIP (Radford et al., 2021) embeddings. An alternative popular strategy involves inverting the input image to a noise vector, and using this noise vector as input (instead of random noise) to the edit-conditioned generation process (Meng et al., 2021). DiffEdit (Couairon et al., 2022b) extends the method with a deterministic inversion strategy and automated edit mask estimation. EDICT (Wallace et al., 2023) further proposes an improved inversion strategy inspired from affine coupling layers. While benefiting from higher flexibility and speed, these methods typically have reduced ability to perform large or complex modifications due to their entirely training-free nature.

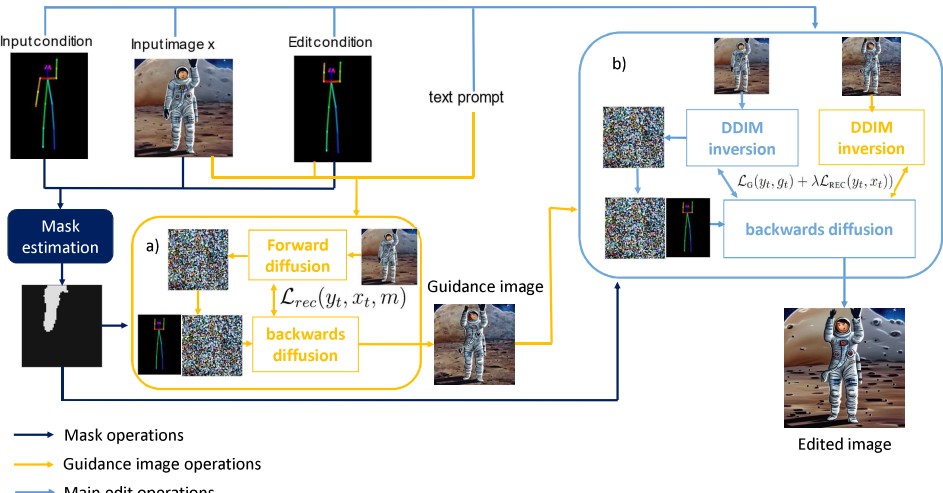

Figure 1: Multi-step overview of our proposed framework, a) guidance image generation block using preservation loss only, b) final image generation block using both preservation and guidance losses.

**Inference-time optimisation** is an intermediate approach that leverages frozen pre-trained models, but introduces an optimisation component at inference-time, typically to update image features. Initial works leveraged CLIP text-image similarity losses to guide the editing process using GANs (Patashnik et al., 2021) or diffusion models (Avrahami et al., 2022; Kim et al., 2022). Mokady et al. (2023) introduce an extension to prompt-to-prompt for natural image editing using a pivotal inversion technique and null-text optimisation. Parmar et al. (2023) aim to preserve edit-text irrelevant content by imposing consistency between the text-image cross-attention maps before and after edit. The method requires computing a textual embedding direction in advance using a thousand sentences, which is expensive and makes extension to additional edit conditions challenging, and is limited by the resolution of the cross-attention map for structure guidance.

These pre-existing approaches have focused on text-driven editing methods, requiring complex strategies to achieve precise local modifications, notably by leveraging attention mechanisms. Epstein et al. (2023) introduce an optimisation procedure to guide the positioning, size and appearance of objects in an image using precise edit instructions (*e.g.* object desired pixel location) and dedicated loss functions. These loss functions drive the update of intermediate image features and attention maps. This approach requires expensive manipulation of attention layers, and explicit estimation of local positioning of objects at all levels of the diffusion UNet. It is heavily dependent on the accuracy of internal attention maps, leading to leakage when instances are not strongly salient. By developing a method capable of leveraging explicit layout constraints, we are able to provide precise edit instructions with much simpler optimisation constraints and remove dependency on internal attention map representations. Furthermore, our disentangling strategy between preservation and modification further affords additional controllability and flexibility, allowing to adjust the influence of each subtask easily.

## 3 METHOD

Given an input image $I$ and a semantic edit condition $C$, our objective is to locally modify $I$ according to $C$ while accurately preserving image content not directly related to the desired edit. We assume access to a text-based image caption $S$, hand-crafted or generated by an image captioning model, and a pre-trained text-to-image diffusion model capable of leveraging image-space layout conditions (*e.g.* pose or scribble). The latter model can be instantiated using *e.g.* ControlNet modules (Zhang & Agrawala, 2023). In contrast to pre-existing work that typically restricts edit condition inputs to text form, we assume $C$ can be provided as either textual or image-space layout inputs.

An overview of our method is provided in Fig. 1. We propose a semantic image editing method by leveraging the concept of inference time optimisation. Our strategy is uniquely designed to

accommodate multiple types of edit instruction including text, pose, scribble and more complex image layouts. We build on the concept of image-to-image editing where an input image is progressively converted into a noise vector, then denoised across multiple timesteps using predictions from a pretrained latent diffusion model, according to edit condition $C$. We introduce an optimisation procedure over latent image features within this encoding-decoding process. We propose to decompose the edit task into two competing subtasks, using distinct content preservation and local image modification losses. Our preservation loss imposes consistency between the input and edited images, thus driving the preservation of original details. The image modification loss provides appearance guidance for the expected image edit area, using an intermediate edited image obtained with reduced content preservation constraints. We additionally enhance our preservation loss with a binary mask that defines a local edit area, thus enabling both content preservation and locally accurate editing.

We provide an overview of diffusion models and image-to-image editing Meng et al. (2021) in Sec. 3.1, followed by description of our core method components in Sec. 3.2: preservation loss, guidance loss, mask, and image guidance generation.

## 3.1 PRELIMINARIES

Denoising diffusion models learn to invert a multi-step diffusion process, where an input image $x_0$ is progressively converted into a Gaussian noise vector $x_T$, by iteratively adding noise $\epsilon \sim \mathcal{N}(0, I)$ across multiple timesteps, according to a specific variance schedule. Given $x_T$ and a text prompt, a model $\epsilon_\theta$ (*e.g.* a UNet (Ronneberger et al., 2015)) is trained to reconstruct the input image by predicting the noise vector to remove at a given timestep $t$. At inference time, $\epsilon_\theta$ iteratively estimates noise vectors across $T$ timesteps to generate a new image from random noise. Using the DDIM deterministic sampling procedure (Song et al., 2020), we can estimate the updated intermediate denoised image at timestep $t-1$ as:

$$x_{t-1} = \sqrt{\frac{\alpha_{t-1}}{\alpha_t}} x_t + \sqrt{\alpha_{t-1}} \left( \sqrt{\frac{1}{\alpha_{t-1}} - 1} - \sqrt{\frac{1}{\alpha_t} - 1} \right) \epsilon_\theta(x_t, t, C), \tag{1}$$

where $C$ is the embedding of the text prompt condition, and $\alpha_t$ is a measure of the noise level, dependent on the variance scheduler.

The forward-backward diffusion process can be viewed as an image encoding-decoding process. The SDEdit algorithm (Meng et al., 2021), commonly referred to as image-to-image, leverages this concept for the purpose of image editing. The key idea is to stop the forward diffusion process at an intermediate timestep $t_E < T$, and reconstruct the image, from $t_E$, using a *different* text condition. This strategy typically results in preservation of some of the original image attributes yet often leads to unwanted global modifications (Couairon et al., 2022b). Leveraging the trained diffusion model $\epsilon_\theta$, in conjunction with a deterministic reverse-DDIM process that encodes the input image (Couairon et al., 2022b), can substantially improve image fidelity, yet sometimes at the cost of edit flexibility. Concretely, we carry out the same process as the backwards diffusion (decoding process), in the $x_0 \rightarrow x_{t_E}$ direction, using the following inverse update:

$$x_{t+1} = \sqrt{\frac{\alpha_{t+1}}{\alpha_t}} x_t + \sqrt{\alpha_{t+1}} \left( \sqrt{\frac{1}{\alpha_{t+1}} - 1} - \sqrt{\frac{1}{\alpha_t} - 1} \right) \epsilon_\theta(x_t, t). \tag{2}$$

This process is commonly referred to as DDIM inversion; a naive way to invert a natural image (*i.e.* estimate the noise vector that will generate this exact image).

We highlight that in this work, we consider diffusion models that operate in a latent space (Rombach et al., 2022), where $x_0 = f_\phi(I)$ is a latent representation of input image $I$, obtained using a pre-trained Variational AutoEncoder (VAE) model.

## 3.2 OPTIMISATION-BASED IMAGE EDITING

**Inference-Time Optimisation: Preservation loss.** Our method builds on the SDEdit process, outlined in Sec. 3.1, to perform image editing. We encode our input latent image $x_0$ with DDIM inversion, up to an intermediate encoding level $t_E$, obtaining intermediate noised image latent $x_{t_E}$. Using this noised latent as input, we decode our image by firstly defining an edit condition in order

to obtain latent edited image $y_0$. This edit condition can be a modified text prompt, or a modified layout input (*e.g.* modified pose, scribble or edge map). We note that multiple edits can be considered *conjointly* (*e.g.* modified pose *and* text prompt), without explicit methodology modifications.

To enforce appearance and structure consistency between the input and edited images, we introduce our latent feature optimisation process. At timestep $t$ of the backwards diffusion process, we update the latent image features $y_t$ to increase similarity with $x_t$. To this end, we introduce a reconstruction task that pushes $y_t$ towards the direction of $x_t$. Formally, we update $y_t$ as:

$$y'_t = y_t - \gamma \nabla_{y_t} \|y_t - x_t\|_2^2, \tag{3}$$

where $\gamma$ is the learning rate. This update is repeated for $k$ gradient updates, pushing $y_t$ towards $x_t$ whilst also avoiding the trivial solution $y'_t = x_t$ through limiting the number of gradient updates. We then proceed with the standard diffusion process, using $y'_t$ as input to the diffusion model at the next timestep. Providing a binary mask $m$ to identify image regions that our edit condition will impact explicitly provides appropriate preservation of related original image content. Integration of mask $m$ into our reconstruction loss also leads to implicitly unconstrained flexibility in image regions that *are* valid edit regions:

$$\mathcal{L}_{\text{REC}}(y_t, x_t, m) = \|m \odot y_t - (1 - m) \odot x_t\|_2^2, \tag{4}$$

*i.e.* the reconstruction loss is only computed within the masked area. State of the art approaches typically invoke $m$, after computing $y_{t-1}$ using Eq. 1, by updating latents as $y_{t-1}^m = m \odot y_{t-1} + (1 - m) \odot x_{t-1}$. We find that by introducing the mask constraint in an optimisation framework, our strategy is more robust to mask quality (*e.g.* underestimation of the edit area) and reduces the risks of introducing artefacts, whilst preserving non-edit regions.

**Inference-Time Optimisation: Guidance loss.** While our preservation loss (Eq. (4)) focuses on faithfully maintaining image content, our complementary guidance loss seeks to improve controllability, enable precise local modification, and steer edited areas towards a desired appearance. We assume availability of an image $G$, with latent representation $g_0$, that accurately depicts the desired appearance of the target edit area. Our solution to obtain $g_0$ will be discussed later in this section. We first encode $g_0$ to obtain $g_{t_E}$, in similar fashion to the encoding process of $x_0$. At timestep $t$, we update $y_t$ towards increasing the cosine similarity with the guidance image features:

$$y'_t = y_t - \gamma \nabla_{y_t} \mathcal{L}_{\text{G}}(y_t, g_t) \qquad \text{where} \qquad \mathcal{L}_{\text{G}} = 1 - \frac{y_t \cdot g_t}{\|y_t\| \|g_t\|}. \tag{5}$$

As larger differences are expected between $y$ and $g$ (*c.f.* between $y$ and $x$), we make use of a more conservative cosine similarity loss for this task, *c.f.* the previous reconstruction MSE loss. By combining guidance and preservation losses, our complete disentangled ITO process can update intermediate image features as:

$$y'_t = y_t - \gamma \nabla_{y_t} ((1 - \lambda) \mathcal{L}_{\text{G}}(y_t, g_t) + \lambda \mathcal{L}_{\text{REC}}(y_t, x_t)), \tag{6}$$

where $\lambda$ is a hyperparameter balancing the influence of the editing subtasks. As such, adjusting $\lambda$ can allow the user to balance between content preservation and edit instruction accuracy, according to preference. We carry out feature updates for the first $t_u$ steps of the backwards diffusion process.

**Guidance Image Generation.** The guidance image plays the crucial role of providing an accurate depiction of the expected appearance of the edited region. This is information that we do not have available beforehand, and need to generate via an intermediate step. We rely on the observation that encoding our input image using random noise (as in (Meng et al., 2021)) facilitates image modifications at the expense of reduced content preservation; while using DDIM inversion (as in Couairon et al. (2022b)) only affords more conservative changes. In contrast with our final edited output, which has to accurately preserve image background, our focus here is on accurately editing the regions of interest. Based on this observation, we generate our guidance image using our ITO process with $\lambda = 1$ (preservation only), with one key difference: here, our input image is encoded using random noise, instead of DDIM inversion to afford larger modifications of the input image.

**Edit Mask Generation.** We estimate the binary edit mask by leveraging the mask generation approach proposed in Couairon et al. (2022b). The idea proposes to measure the differences between noise estimates using (1) the original image text prompt and (2) the edit prompt as conditioning, effectively inferring which image regions are affected most by the differing conditions. In contrast with Couairon et al. (2022b), which only considers text conditions, we additionally estimate masks

from layout conditions. After encoding our input $x_0$ using random noise with seed $s$, we obtain $x_E(s)$ and carry out the backward diffusion $x_E(s) \rightarrow y_0(\mathcal{C}, s)$ for two different conditions: the original condition $\mathcal{C}_O$ (*e.g.* original prompt or pose), and the edit condition $\mathcal{C}_{EDIT}$. In settings where only a layout condition is used, both images share the same text prompt and the layout input is integrated in the diffusion process using conditioning modules such as ControlNet (Zhang & Agrawala, 2023) or T2IAdapter (Mou et al., 2023). We estimate our mask by comparing noise estimates at the last timestep:

$$m(x_0, \mathcal{C}_{EDIT}) = \frac{1}{n} \sum_{i \in [1:n]} |\epsilon_\theta(x_1, 1, \mathcal{C}_{EDIT}, s_i) - \epsilon_\theta(x_1, 1, \mathcal{C}_O, s_i)|, \quad (7)$$

where $m$ is then converted to a binary mask using threshold $\tau$. The mask is averaged over $n$ runs with different seeds to increase the stability and accuracy of noise estimates.

**Task Difficulty Driven Modular Process.** Disentangling preservation from modification provides a modular edit procedure, where a user can adapt constraints according to preferences and task difficulty; by adjusting loss-weighting parameter $\lambda$. We illustrate our complete edit procedure in Fig. 1, which comprises three steps: mask estimation, guidance image generation, and image editing. The role of the guidance image is to ensure layout constraints are respected. This is typically useful for difficult pose modifications, where the ControlNet conditioned image-to-image setting fails to achieve the desired modification. For simple edit tasks (*e.g.* small modifications such as a 90 degrees arm movement, uniform backgrounds), it is possible to set $\lambda = 1$, effectively focusing solely on the preservation task. This skips the guidance image generation step (depicted in yellow in Fig. 1), simplifying the edit process. We demonstrate empirically how this simpler setting has very similar behaviour to DiffEdit (Couairon et al., 2022b).

## 4 EXPERIMENTS

In this section, we describe our experimental setup, followed by experiments on different conditions.

**Implementation.** In our experiments we use ControlNet (Zhang & Agrawala, 2023) to control a Stable Diffusion model v1.5 (Rombach et al., 2022) with layout edit conditions. We use 25 sampling steps, and an encoding ratio of 1 ($t_E = 25$) unless specified otherwise. Balancing parameter $\lambda$ is set to 0.6 unless specified otherwise. For mask generation, we follow the parameters of (Couairon et al., 2022b), but introduce Gaussian smoothing of the mask, and use a threshold of 0.1. When using a mixture of guidance and preservation loss, we optimise features for the first $t_u = 15$ timesteps for $k = 1$ gradient step, for both guidance generation and final editing steps. We use the ADAM optimiser with a learning rate $\gamma = 0.1$.

**Layout-conditioned image editing.** As pre-existing image editing methods have not been developed for layout conditioning, we use ControlNet as our simplest baseline. Additionally, we compare our method to a combination of ControlNet and DiffEdit (Couairon et al., 2022b) using the edit mask generated by our method. In this setting, we set the encoding ratio for DiffEdit to 1.0, as we have observed better results, and use the binary mask estimated using our updated approach for layout conditions. Figs. 2 and 3 show visual results for pose and scribble conditions, where we edit an image originally generated using a ControlNet pose or scribble constraint. Fig. 2 shows visual results under different pose conditions. We can see that our approach is the only one to consistently achieve the correct pose modification while preserving image content. As expected, ControlNet successfully generates images with the correct pose, but fails to preserve image content. In contrast, while DiffEdit preserves image content thanks to its masking process, it often struggles to achieve the correct pose change, especially for more difficult queries. We highlight in particular how DiffEdit often suffers from local artefacts, often due to the strict mask blending constraint. Lastly, we highlight DiffEdit's performance is very close to our full preservation special case ($\lambda = 1$), with ours having fewer image artifacts (see the altered image quality for the dancer and bear images). This shows how our preservation loss provides a robust alternative to DiffEdit's mask merging, and the benefits of using our guidance image. Our scribble-based editing results, shown in Fig. 3, show a similar behaviour.

**Text-driven image editing.** Quantitative evaluation protocols and metrics, as well as pre-existing methods, only consider text-guided editing. As a result, we restrict our quantitative evaluation to text conditioning only. We follow the evaluation protocol of DiffEdit (Couairon et al., 2022b) on the ImageNet (Deng et al., 2009) dataset. The task is to edit an image belonging to one class, such that it

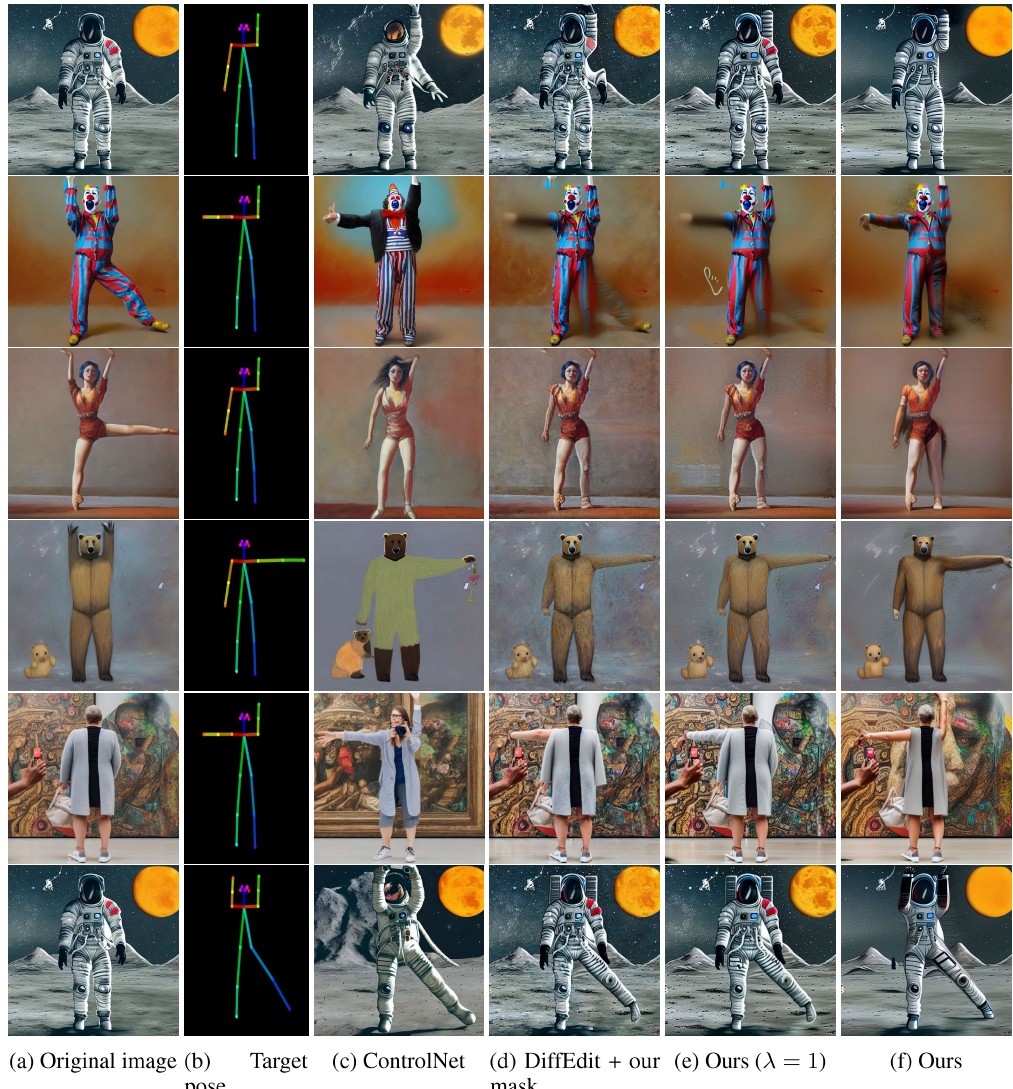

    (a) Original image  (b)    Target    (c) ControlNet   (d) DiffEdit + our  (e) Ours ($\lambda = 1$)    (f) Ours
                  pose                                    mask

Figure 2: Visual results for pose based editing.

depicts an object belonging to another class. This is indicated by an edit text prompt comprising the name of the new class. Given the nature of the ImageNet dataset, edits generally require modifying the main object in the scene.

We compare to our results to state of the art methods with default parameters: Prompt-to-prompt Hertz et al. (2022) and DiffEdit Couairon et al. (2022b) (encoding ratio=0.7). We evaluate performance from four angles: 1) image preservation by computing the L1 distance between original and edited images, 2) image quality by computing the CSFID score (Couairon et al., 2022a), which is a class-conditional FID metric (Heusel et al., 2017) measuring both image realism and consistency with respect to the transformation prompt. 3) Quality of the semantic modification, by measuring classification accuracy of the edited image using a) the new category as ground truth, and b) the original category as ground truth, and finally 4) we measure relative image quality and prompt faithfulness using the Pickscore (Kirstain et al., 2023). As shown in Table 1, our approach outperforms competing works for all metrics, except the L1 consistency metric. This can be expected as edited regions often are the only subject of the image, such that successful edits achieve more substantial image modifications.

Additionally, Fig. 4 shows visual results using text-based editing. Here, we edit images generated with a Stable Diffusion v1.5 model (Rombach et al., 2022). We provide DiffEdit's results for an

| Metric | DiffEdit | Prompt-to-prompt | Ours |
|---|---|---|---|
| L1 distance ↓ | **0.124** | 0.128 | 0.136 |
| CSFID ↓ | 67.7 | 74.2 | **57.2** |
| Classification accuracy after edit (as new category) ↑ | 37.8 | 34.2 | **60.1** |
| Classification accuracy after edit (as original category) ↓ | 45.9 | 45.5 | **20.0** |
| Pickscore ↑ | 0.31 | 0.27 | **0.41** |

Table 1: Quantitative comparison of text-guided image editing on the ImageNet dataset.

encoding ratio of 0.7 and 1.0, highlighting the strong similarity between our method when only using the preservation loss ($\lambda = 1$). As evidenced with the zebra image, where DiffEdit shows local artifacts, we are less sensitive to subpar editing masks. We note how our method can achieve successful conversions, such as the bowl of grapes, where pre-existing works fail.

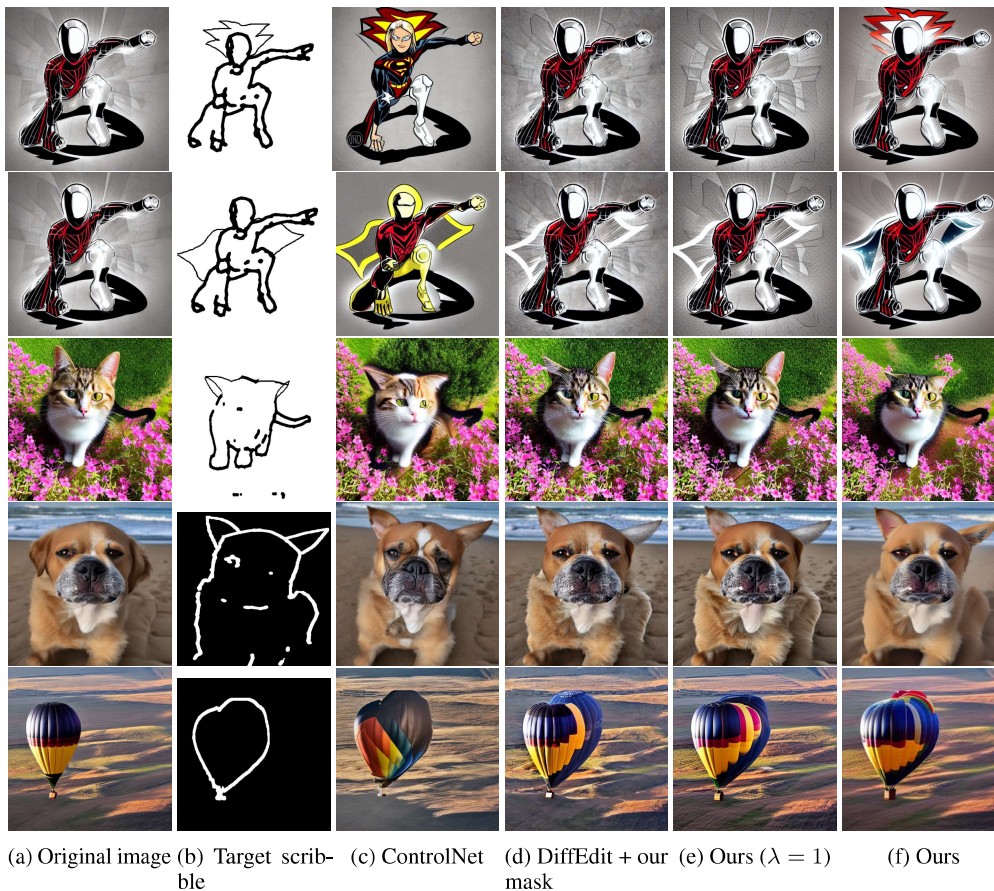

(a) Original image  (b) Target scribble  (c) ControlNet  (d) DiffEdit + our mask  (e) Ours ($\lambda = 1$)  (f) Ours

Figure 3: Visual results for scribble based editing.

**Parameter Study.** In Fig. 5, we illustrate the impact of adjusting balancing parameter $\lambda$ which controls the relative influence of preservation and guidance losses. We can see that lower values of $\lambda$ focus on foreground and accurate positioning. Larger $\lambda$ values increase background details at the cost of additional artifacts when $\lambda$ is too high. Extreme values yield an output very close to the guidance image ($\lambda = 0$), or very similar to the DiffEdit performance ($\lambda = 1$, high background fidelity, but poorer pose edit quality). This behaviour highlights how, for simpler edit instructions (e.g. the bear or dancer in Fig. 2, where DiffEdit successfully achieves the pose modification), one can discard the guidance image component and edit the image with a simple preservation loss. More evidence of this is available in the supplementary materials.

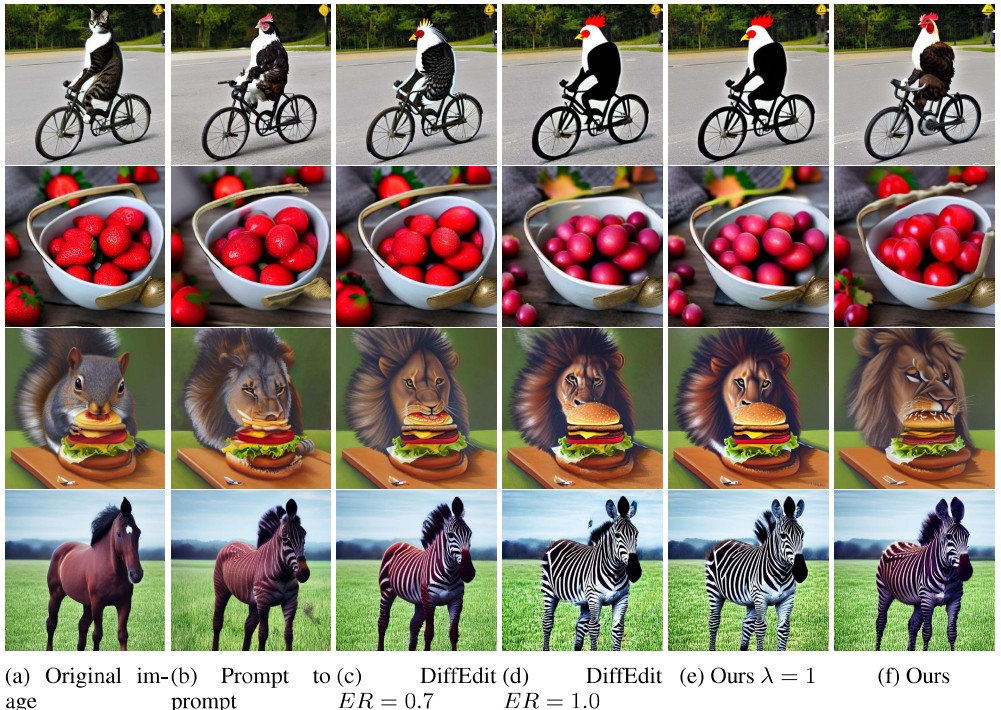

(a) Original image (b) Prompt to prompt (c) DiffEdit $ER = 0.7$ (d) DiffEdit $ER = 1.0$ (e) Ours $\lambda = 1$ (f) Ours

Figure 4: Visual results for text-conditioned image editing. From top to bottom, input/edit prompts are: a cat/chicken riding a bicycle, a bowl full of strawberries/grapes, a squirrel/lion eating a burger, a horse/zebra in a field.

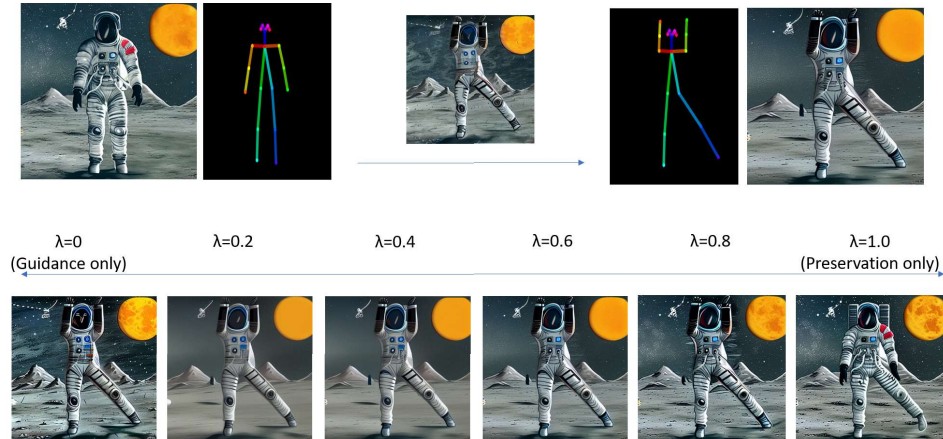

Figure 5: Influence of balancing parameter $\lambda$. Top left: input image and input pose, middle: guidance image, top right: output image and target pose.

## 5 CONCLUSION AND LIMITATIONS

We propose a novel method for diffusion model-driven image editing that is uniquely designed to leverage layout-based edit conditions, in addition to text instructions. By disentangling preservation and modification in two separately controllable functions, we build a flexible solution that allows the user to adjust the editing output according to preference. Key limitations involve reduced preservation ability in certain parameter settings, inconsistent mask quality, and the influence of the guidance image to achieve the desired modification, where a poor quality image can alter output quality. Future work will explore alternative mask and guidance generation methods, potentially investigating different image inversion strategies.

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

# A SUPPLEMENTARY

## A.1 INFERENCE-TIME OPTIMISATION: DETAILED ILLUSTRATION

In Fig. 6 we provide a detailed visualisation of our ITO process with guidance and preservation losses, so as to facilitate user comprehension.

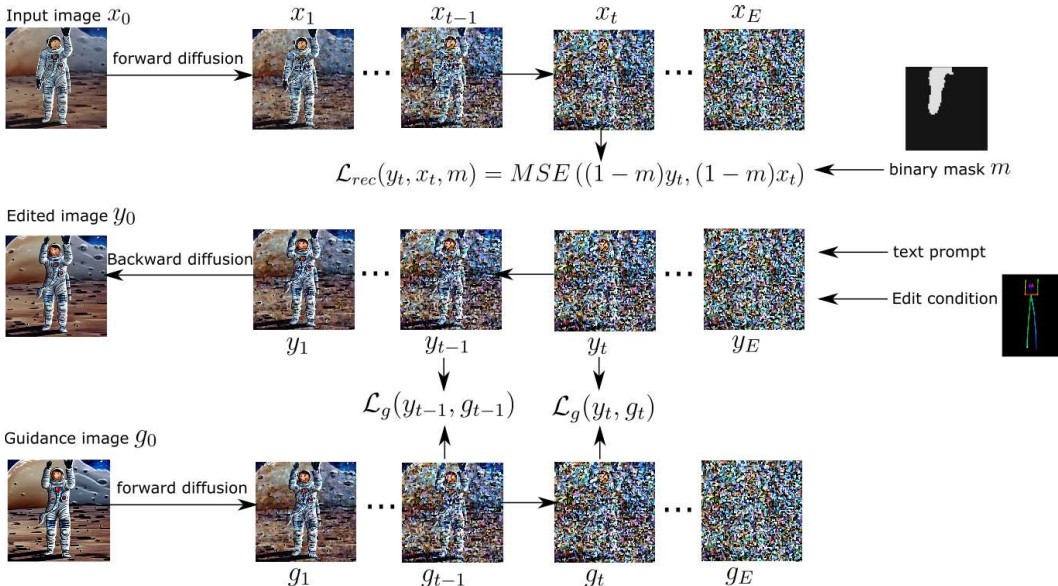

Figure 6: Overview of our inference time optimisation method with preservation and guidance losses.

## A.2 PARAMETER STUDY: UPDATE STEPS

In Fig. 7, we visualise the influence of parameters $t_u$ and $k$ over both guidance image generation and final edit. We provide visual results for two image edits using pose conditioning for $t_u \in \{5, 10, 15, 20, 25\}$ and $k = \{1, 30\}$, choosing images with complex backgrounds to highlight the impact of these parameters more clearly. We can see that $t_u$ has a larger impact over the generated images than $k$, especially with regards to the guidance image. Compared to the input image, we can see that the best quality guidance images (in terms of content preservation, achieved modification, and introduced artefacts) is obtained for $t_u = 20, k = 30$, while $t_u = 15, k = 1$ yields a better final edit. For guidance images, better content preservation at $t_u = 20$ is notably evidenced by the woman's hair in Fig. 7e, and the symbol on the top left of the astronaut images in Fig. 7b. Increasing $k$ has a minor impact, but yields slightly improved results (see the sleeve on the woman's right shoulder in Fig. 7e). For final edits (Figures 7c and 7f), $t_u = 15, k = 1$ yields noticeably better results, with reduced artefacts (see astronaut image, both images for $t_u = 25$ and all settings where $k = 30$). In particular, we can see that noticeable artefacts are introduced within the masked area for lower values of $t_u$ and $k > 1$.

For the sake of consistency and speed, we use $t_u = 15, k = 1$ in all settings in our experiments, but note here results could be further improved by adjusting guidance image parameters to $t_u = 20, k = 30$. We further highlight that the optimal value of $t_u$ is dependent on the number of sampling steps used to generate the image (here we use 25 steps, as discussed in section 4 of the main paper).

## A.3 VISUALISATION OF INTERMEDIATE EDIT STEPS

In Fig. 8, we provide detailed visualisations of results presented in the main paper, more specifically showing intermediate outputs such as guidance images and estimated edit mask. We can see that our mask successfully identifies regions modified by the layout change. We also highlight how the

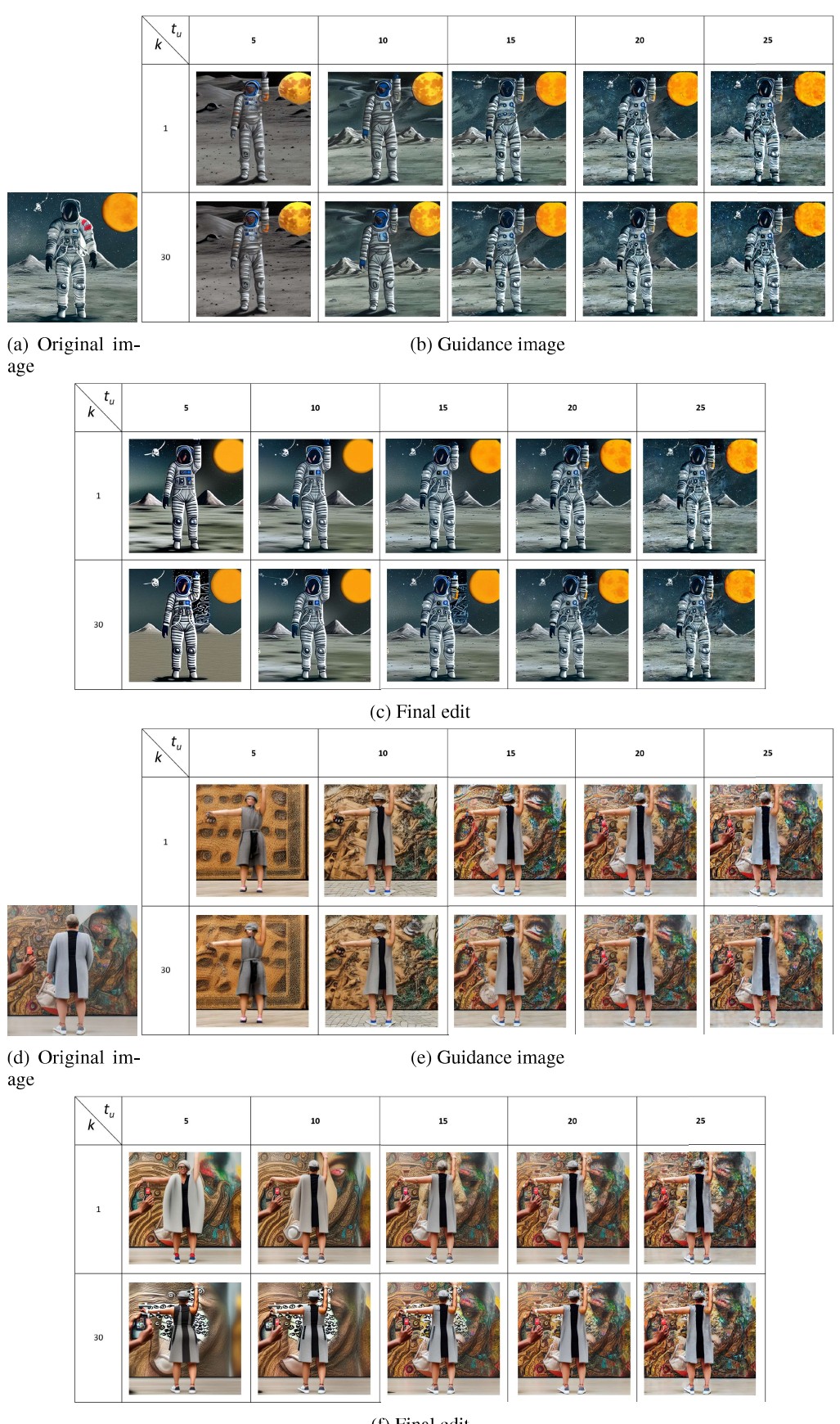

Figure 7: Impact of parameters $t_u$ and $k$ on guidance image and final edit generation for two example images.

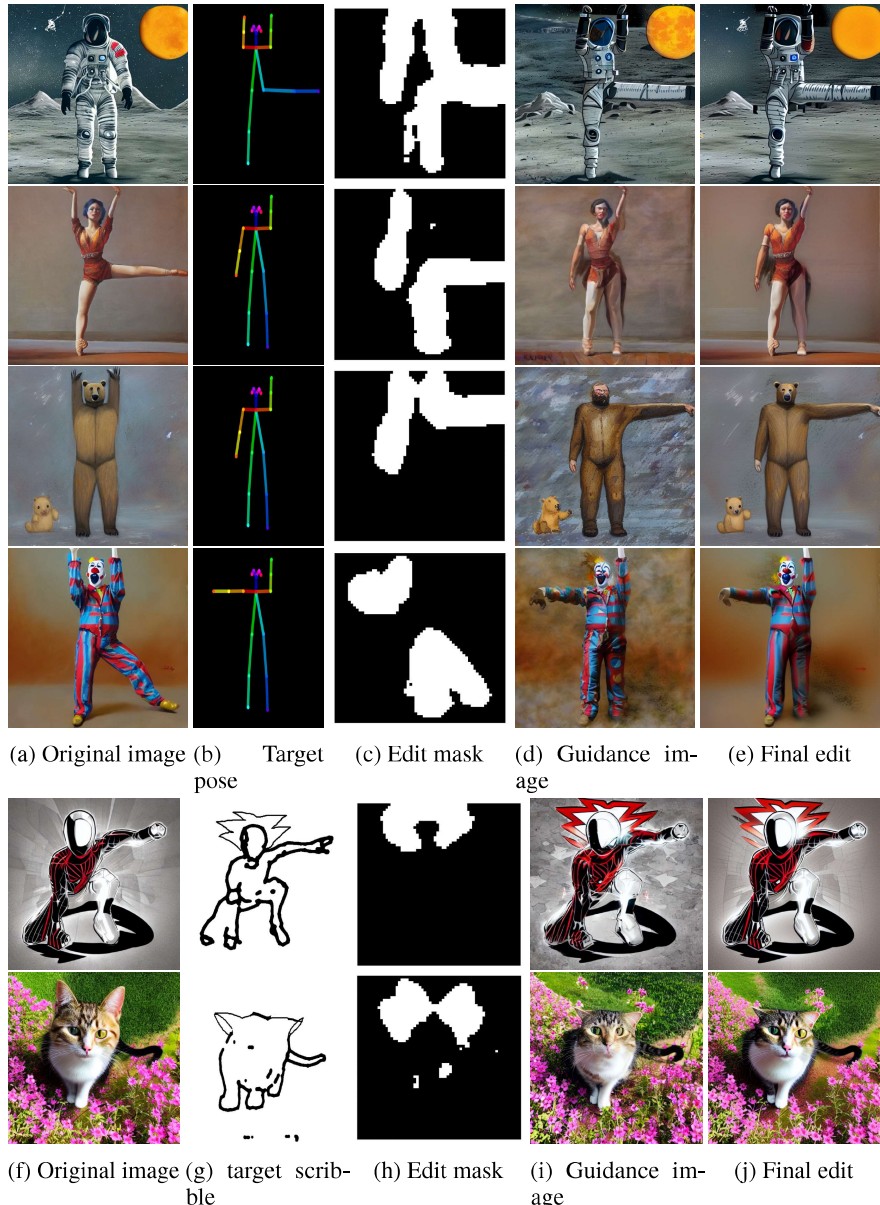

(a) Original image (b) Target pose (c) Edit mask (d) Guidance image (e) Final edit

(f) Original image (g) target scribble (h) Edit mask (i) Guidance image (j) Final edit

Figure 8: Visualisation of intermediate edit outputs on multiple pose and scribble conditionned image modifications.

guidance image has reduced content preservation, and how larger changes are observed when the masked area is larger. This suggests that task difficulty could be estimated based on mask size.

## A.4  IMAGENET EDITING: VISUAL EXAMPLES

In Fig. 9, we provide a visual comparison of edits carried out on the ImageNet dataset, i.e. our text-editing quantitative experiments reported in the main paper. We highlight how our method's flexibility allows to achieve larger modifications (i.e. shaker to coffee mug, english springer to golden retriever) when other methods fail, while preserving image structure. We further note how masks that are too restrictive can negatively impact DiffEdit based editing (see e.g. coyote to red fox where the mask is limited to the animal's head). While we leverage the same mask as DiffEdit, its integration into our optimisation procedure affords more flexibility.

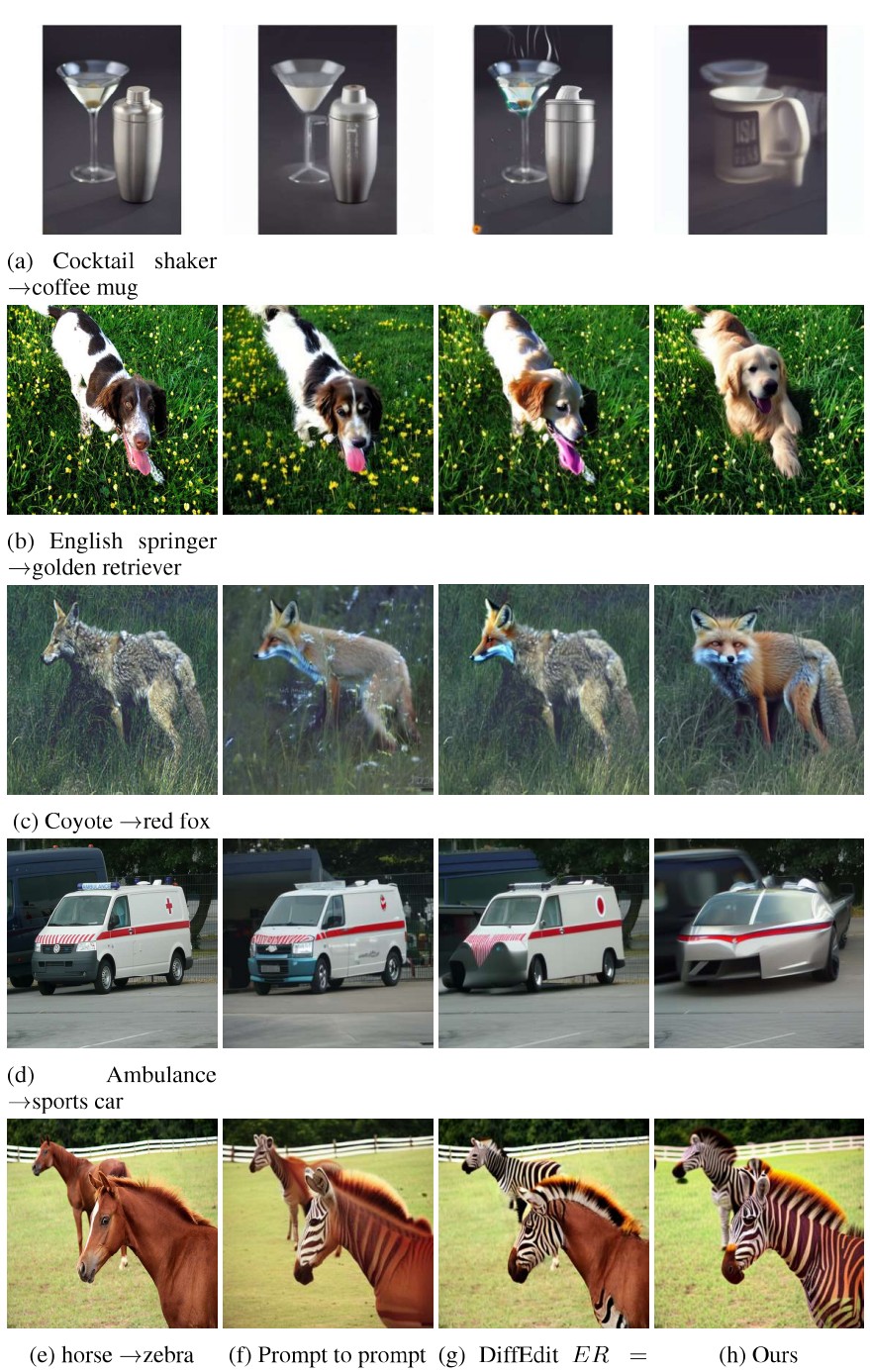

(a) Cocktail shaker →coffee mug

(b) English springer →golden retriever

(c) Coyote →red fox

(d) Ambulance →sports car

(e) horse →zebra      (f) Prompt to prompt  (g) DiffEdit $ER = 0.7$      (h) Ours

Figure 9: Visual results of the ImageNet editing task.

### A.4.1 ADDITIONAL EDITING SETTINGS

In this section, we provide additional results depicting our model's ability to perform editing with different types of edit instructions. We demonstrate image editing using Hough line map conditions, as well as multi-condition editing.

Fig. 10 shows an example edit using Hough line maps as conditioning, compared to DiffEdit. As with other forms of conditioning, we can see the DiffEdit output is very similar to our $\lambda = 1$ output, while our $\lambda = 0.6$ output allows more larger modifications more aligned with the new conditioning.

Fig. 11 shows examples combining multiple types of editing conditions, namely text+scribble and text+ pose. We compare our results to ControlNet and DiffEdit. We can see that both our approach and DiffEdit are able to integrate two types of edit conditions, while ControlNet fails to maintain visual consistency, as one would expect from the lack of preservation mechanisms.

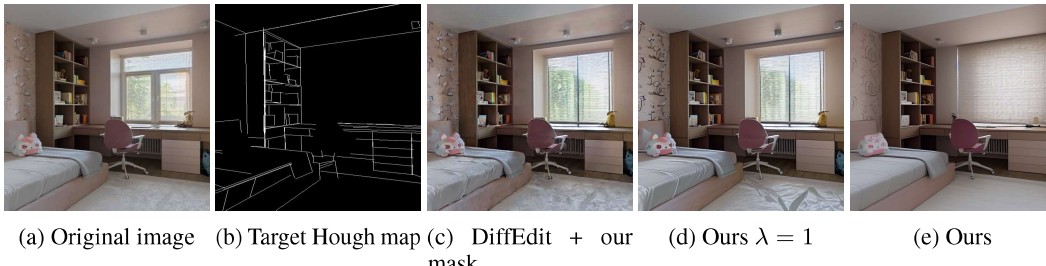

(a) Original image   (b) Target Hough map   (c) DiffEdit + our mask   (d) Ours $\lambda = 1$   (e) Ours

Figure 10: Visual results of editing with Hough line map conditioning.

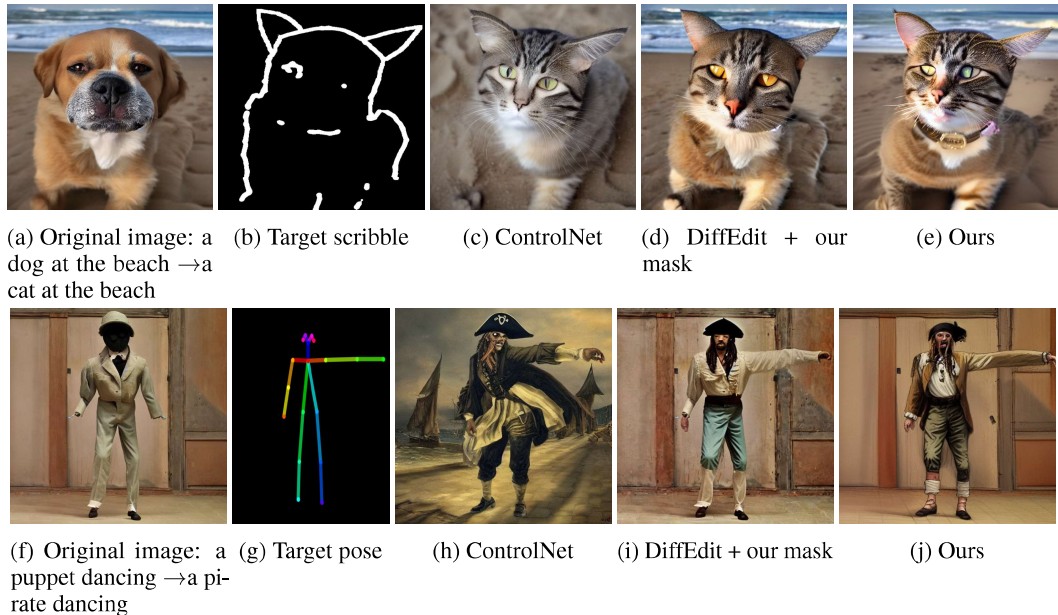

(a) Original image: a dog at the beach →a cat at the beach   (b) Target scribble   (c) ControlNet   (d) DiffEdit + our mask   (e) Ours

(f) Original image: a puppet dancing →a pirate dancing   (g) Target pose   (h) ControlNet   (i) DiffEdit + our mask   (j) Ours

Figure 11: Visual results for multi-condition editing, combining pose/scribble with text edit instructions.

### A.5 ADDRESSING FAILURE MODES WITH GUIDANCE IMAGE REFINEMENT

For certain images with complex background and large editing constraints, it can be difficult to generate a guidance image that successfully preserves all image and background details. This can lead to the introduction of local image artefacts. To address this limitation, we investigate whether it is possible to refine our guidance image $G$ using information from our original input image $I$, effectively re-injecting information from the source image. To do this, we leverage our inference time optimisation with $\lambda = 0$, leveraging only a guidance loss. In this setting, the input image to be updated is $G$, and the reference image is $I$. We carry out this update using $t_u = 6$ and $k = 1$. We note that this approach can alter the quality of the guidance image, as it is encoded/decoded multiple times, and therefore recommend integrating this step only if editing results are unsatisfactory with our original architecture.

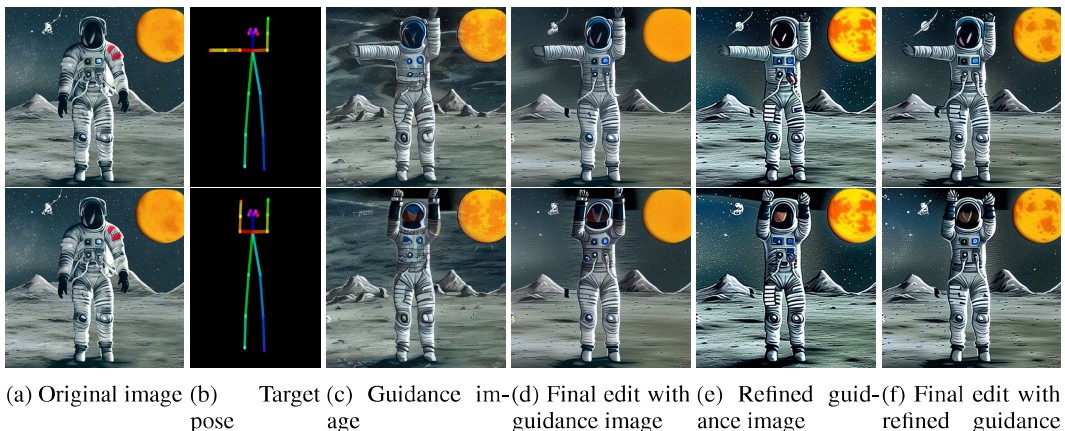

(a) Original image (b) Target pose (c) Guidance image (d) Final edit with guidance image (e) Refined guidance image (f) Final edit with refined guidance image

Figure 12: Illustration of the impact of our additional guidance image refinement process, using a pose editing condition.

We illustrate the failure mode and proposed guidance image refinement solution in Fig 12, showing how this additional step can successfully eliminate artefacts introduced by our original guidance image.

