# OpenReview forum: "Optimisation-Based Multi-Modal Semantic Image Editing"
_ICLR.cc/2024/Conference — ICLR 2024 Conference Withdrawn Submission_

### Official Review · Reviewer_5aGY · 2023-10-28

**Soundness:** 2 fair
**Presentation:** 2 fair
**Contribution:** 2 fair
**Rating:** 3
**Confidence:** 3

**Summary:**

This paper proposes a new optimization-based method to edit images generated by diffusion models. The key idea is to first generate a guidance image satisfying the input edit condition using a combination of (Meng et al., 2021) and DDIM inversion, and then find a balance between the trade-off of preservation (the original image) and guidance image during the image generation process.

**Strengths:**

- The main ideas presented are simple and easy to understand.

**Weaknesses:**

In general, I feel this paper needs some more work to be acceptable:
- The results are not satisfactory and it is difficult to tell whether it really improves upon previous works. Specifically, many of the generated images are overly smoothed and contain fewer details than the original or the guidance images; most of the results are good in some aspects and worse in others. I suspect the reason comes from the naive loss function used (Eq. 6), where the two objects are competing and may not yield a satisfactory balance.
- There is not much technical novelty as the methods proposed are either common practice or a simple combination of previous methods. 1) The idea of preservation vs. guidance and the losses are common practice and not novel. It would be fine if it yields surprisingly good results but as mentioned above, this seems not the case. 2) The generation of guidance images is not new either, which is a combination of the random noise used in (Meng et al., 2021) and DDIM inversion.
- The necessity of the preservation vs. guidance optimization is not justified. For example, in Fig. 5 top row, the guidance image has more details (although a bit different from the input) but the optimized image is overly smoothed. The guidance image alone might be preferable.
- The English is understandable but the presentation can be further polished to facilitate understanding, especially the novelty of this paper.

Minor:
- Eq. 4 is confusing, why $m$ is used for $y_t$ and $(1-m)$ is used for $x_t$? This contradicts the MSE loss in supplementary materials.
- What is the difference between $L_{rec}$ and $L_{REC}$ in Fig. 1?

**Questions:**

Please see weaknesses above.

---

### Official Review · Reviewer_aHvV · 2023-10-31

**Soundness:** 4 excellent
**Presentation:** 3 good
**Contribution:** 3 good
**Rating:** 5
**Confidence:** 4

**Summary:**

This paper proposes a new inference-time optimization method for diffusion-based image editing. More specifically, the authors propose a reconstruction loss and guidance loss. The results are promising.

**Strengths:**

1. This paper proposes a new method for inference-time optimizaion based image editing, which is capable of handling image layout condition and can better preserve background for local editing tasks. The main idea is reaonable.

2. The comparisons are comprehensive and the results are promising.

3. The paper writing is clear and easy to follow.

**Weaknesses:**

1. My main concern is the novelty part. Since the main process is based on SDEdit, and the proposed edit mask and guidance image are similar or based on the results of DDIM Inversion. Then it seems that the only contribution is combinaing them via two proposed losses?

2. What is the main difference between guidance image and final result? Similar process but which different intial noise to incorporate more diversity?
Does this implicitly mean the upperbound of samantic editing accuracy is this guidance image? And other improvements are just for background preservation?

3. Is this method suitable for real world images rather than editing a generated image?

**Questions:**

Please address the questions in weakness part.

---

### Official Review · Reviewer_Y78e · 2023-10-31

**Soundness:** 3 good
**Presentation:** 3 good
**Contribution:** 3 good
**Rating:** 5
**Confidence:** 2

**Summary:**

The method proposes a  test-time optimization-based image editing model. The paper argues that as opposed to text-based editing methods that rely on diffusion models and limit the precision and accuracy of the edits, their method supports input of various instructions like layouts, pose, scribbles, etc. This provides more flexibility to the edits. To achieve this, the paper employs two different loss functions. The first loss function makes sure the local modification made by the input is respected by the output, while the second one makes sure that the global consistency is maintained given the original image. By using different weights of these losses, the method is able to control the generation process of the edits and hence produce diverse edits according to the user preference. The paper shows the results of manipulation using different inputs throughout the paper, These results are compared with competing methods like ControlNet. Qualitatively, the paper shows how global consistency and precision of edits are maintained using the proposed method.

**Strengths:**

1) The paper uses test-time optimization to approach the task of image editing using diffusion models. This enables the method to support various inputs like edges, poses, layouts, etc., in addition to the text inputs. These inputs help to achieve more precise edited outputs, which are controllable.

2) The losses proposed by the method make sure the precision of the edit and the global consistency of the original image is maintained. The results compared to the competing methods like ControlNet show that results respect the inputs.

3) The paper ablates the choice of the loss function by varying the weights. The results show that different levels of control can be achieved by varying these weights, and suitable configurations can be chosen according to user preference.

**Weaknesses:**

1) The method's evaluation primarily focuses on text-guided image editing, which does not fully represent its capabilities. A quantitative analysis that includes a clear comparison with competing methods, specifically in terms of attributes like edges, segmentation masks, and pose, is absent. Such an analysis would provide a more comprehensive understanding of the method's performance.

2) The paper demonstrates results with only one centered subject. The method is designed to work without relying on large labeled datasets during inference. There are no challenging cases shown to assess the robustness of the method compared to the competing methods.

3) The paper lacks examples featuring multiple objects within a single image. To gain a more comprehensive view of the method's performance, it would be beneficial to showcase how the method handles images with multiple objects and explore any potential failure cases in such scenarios.

4) It appears that the quality of the mask input is a crucial factor for the method's performance. How does the method's performance vary based on the quality of the mask input?

**Questions:**

1) Are there plans to provide quantitative analysis and comparative assessment against competing methods, particularly concerning attributes like edges, segmentation masks, and pose?

2) Could you elaborate on how the method performs when dealing with objects that are not centered in the image, and what considerations or limitations may apply in such scenarios?

3)  Could you discuss how the method handles images with multiple objects and whether there are any potential failure cases in such scenarios?

4)  Can you provide insights into how the method's performance varies based on the quality of the mask input?

---

### Official Review · Reviewer_KWjJ · 2023-11-01

**Soundness:** 2 fair
**Presentation:** 3 good
**Contribution:** 3 good
**Rating:** 6
**Confidence:** 3

**Summary:**

This paper proposes a novel approach to an optimization-based image editing approach that could be controlled through different types of input conditions. They disentangle the editing task into two tasks including preserving the unedited background area and foreground area editing with preservation loss and guidance loss.

**Strengths:**

1. A novel approach for diffusion model editing
2. Most of the quantitative results are better than state-of-the-arts
3. From the qualitative results, the proposed method shows better performance compared to other state-of-the-arts mehtods.

**Weaknesses:**

1. The quantitative evaluation is limited. It would be better if the authors could include ControlNet for quantitative comparison. Also, probably could include
2.  For the L1 distance calculation, it's not so reasonable to take the entire original image and edited image for calculation. It would be better if you applied the mask to calculate only the background area for L1 distance.
3. Some errors (equation 4) or parts not so clear in the paper

**Questions:**

1. For reconstruction loss, it should be only computed within the masked area. However, I don't understand why y_t and x_t are applied with different masked area (x_t:m, y_t:1-m) in equation (4). Could you please explain more about this?
2. For the L1 distance in table 1, how do you calculate it? Do you take the original image and edited image for L1 distance calculation, or do you apply the mask to calculate only the background area for reconstruction comparison?